# Ranking LLM-Generated Loop Invariants for Program Verification

**Saikat Chakraborty, Shuvendu K. Lahiri, Sarah Fakhoury, Madanlal Musuvathi, Akash Lal, Aseem Rastogi, Aditya Senthilnathan, Rahul Sharma, Nikhil Swamy**
Microsoft Research
{saikatc, shuvendu, sfakhoury, madanm, akashl, aseemr, t-adityas, rahsha, nswamy}@microsoft.com

## Abstract

Synthesizing inductive loop invariants is fundamental to automating program verification. In this work, we observe that Large Language Models (such as `gpt-3.5` or `gpt-4`) are capable of synthesizing loop invariants for a class of programs in a 0-shot setting, yet require several samples to generate the correct invariants. This can lead to a large number of calls to a program verifier to establish an invariant. To address this issue, we propose a *re-ranking* approach for the generated results of LLMs. We have designed a ranker that can distinguish between correct inductive invariants and incorrect attempts based on the problem definition. The ranker is optimized as a contrastive ranker. Experimental results demonstrate that this re-ranking mechanism significantly improves the ranking of correct invariants among the generated candidates, leading to a notable reduction in the number of calls to a verifier.

## 1 Introduction

Program verification is a crucial step toward building trustworthy software. Unfortunately, the problem of verifying properties of programs containing loops is undecidable. Verifying properties of programs containing loops boils down to inferring *loop invariants*, which are facts that hold for any iteration of the loop, and also ensure the desired property. There is a rich body of prior work on synthesizing loop invariants for program verification through symbolic techniques (Cousot and Cousot, 1977; Colón et al., 2003; Graf and Saïdi, 1997; McMillan, 2003), and their use in verifying safety properties of real-world programs (Ball et al., 2001; Blanchet et al., 2003; Lahiri et al., 2009). More recently, there is a growing interest in the application of machine learning towards invariant synthesis (Garg et al., 2016; Padhi et al., 2016; Yao et al., 2020; Si et al., 2018).

In recent years, Large Language Models (LLMs) (Radford et al., 2018) have emerged as foundational AI models that have revolutionized Language Processing applications. Though LLMs were originally proposed for natural languages, they have exhibited great success in formal languages such as programming languages (Chen et al., 2021). In fact, with the increased size, models have started to exhibit emergent properties. For example, modern LLMs such as `gpt-3.5` (Ouyang et al., 2022), `gpt-4` (OpenAI, 2023), `PaLM` (Chowdhery et al., 2022) are capable of reasoning about a given task with few-shot (Brown et al., 2020), or even zero-shot prompting (Kojima et al., 2022). Such an impressive footprint of LLM naturally raises the question: *How well can LLMs automatically synthesize inductive loop invariants?*

To this end, we employ two different state-of-the-art LLMs for synthesizing loop invariants. We observe that these models can generate well-formed invariants, but finding the correct one often requires a large number of samples. A solution based on *guess and check*, with the aid of an automated program verifier based on Z3 (De Moura and Bjørner, 2008), can be computationally very expensive due to several invocations on incorrect invariants. To minimize such costs, we propose reranking the generated invariants based on their likelihood of successful verification. Inspired by the use of contrastive learning in information retrieval (Karpukhin et al., 2020), our approach, called *iRank*, transforms the problem and invariants to bring the correct solution closer in vector space while pushing away incorrect ones. Empirical results show that such re-ranking moves the median rank of the verified invariant to 4 in contrast to the expected median rank of 31 for the generations from `gpt-3.5`.

In summary, in this paper, we propose to rerank the LLM-generated loop invariants to reduce the cost of wasted verification effort. We have designed a ranker to contrast correct and incorrect invariants and show a significant reduction in the

invariant checking effort compared to raw LLM generations.

## 2 Related Work

Prior works on loop invariant generation can be broadly grouped into *symbolic* or *machine learning* based. Symbolic approaches either construct invariants that are correct by construction (Cousot and Cousot, 1977; Colón et al., 2003), or leverage Satisfiability-Modulo-Theories (SMT) solvers such as Z3 (De Moura and Bjørner, 2008) to enumerate and check candidate invariants over a space of pre-defined predicates (Flanagan and Leino, 2001; Flanagan and Qadeer, 2002; Lahiri and Bryant, 2007; Gulwani et al., 2009; Fedyukovich and Bodík, 2018) or predicates constructed through variants of Craig's interpolants (McMillan, 2003; Henzinger et al., 2004; Dillig et al., 2013). On the other hand, recent techniques leverage machine learning to synthesize candidate invariants that are checked for correctness using an SMT-based program verifier. Techniques range from incorporating the feedback from a verifier using *active learning* over decision trees (Garg et al., 2016), learning from counter examples (Sharma and Aiken, 2016; Padhi et al., 2016), reinforcement learning over graph neural networks (Si et al., 2018) and the use of continuous logic networks (Yao et al., 2020; Ryan et al., 2020). Unlike these techniques, our approach leverages an LLM for generation and ranks using a purely *neural* model and does not require a program verifier at the inference time. This is important for scenarios where the verifier is semi-automated, as is the case of most real-world program verification tools such as Dafny (Leino, 2010) and F* (Swamy et al., 2011). Finally, Pei et al. (2023) predict program invariants using LLMs, but they do not aim at generating inductive invariants that are sufficient for formal program verification.

## 3 Background & Motivation

### 3.1 Background: Loop Invariant Inference

In this section, we recall the problem of loop invariant generation in program verification. First, let us define a grammar for program statements $S$, integral expressions $a$ and Boolean expressions $b$, operating on scalar variables. Most statements and expressions are self-explanatory.

$$
\begin{array}{rcl}
S & ::= & x := a \mid \textbf{skip} \mid S; S \mid \textbf{if } b \textbf{ then } S \textbf{ else } S \\
a & ::= & n \mid x \mid a + a \mid a - a \mid a * a \mid \ldots \\
b & ::= & \textbf{true} \mid \textbf{false} \mid a = a \mid a < a \mid b \wedge b \mid b \vee b \mid \neg b
\end{array}
$$

In its simplest form, the goal of program verification is to verify that a program fragment satisfies its specifications denoted by the *Hoare-triple* (Hoare, 1969) - $\{pre\}$ **while** $b$ **do** $S$ $\{post\}$. Given a program $p$ and a pair of Boolean expressions (denoted by $b$ in the grammar) $\phi$ and $\psi$ denoting the precondition and postcondition of a program $p$, the Hoare-triple $\{\phi\}$ $p$ $\{\psi\}$ denotes that every terminating execution of $p$ that starts in an *pre*-state satisfying the predicate $\phi$ ends up in a *post*-state that satisfies the predicate $\psi$. Since loops can execute an unbounded number of iterations, verifying programs with a loop requires a loop invariant $i$ that satisfies the following conditions:

$$
\begin{array}{c}
\{pre\} \textbf{ skip } \{i\} \\
\{i \wedge b\} \, S \, \{i\} \\
\{i \wedge \neg b\} \textbf{ skip } \{post\}
\end{array} \quad (1)
$$

The conditions respectively denote that the loop invariant $i$ holds on loop-entry, is preserved by an arbitrary iteration of the loop and implies the post condition on exit. The problem of *loop invariant inference* is to infer an $i$ that satisfies the three checks above, and denoted as $i \vdash p$.

Furthermore, for the loop-free statements $S$ in the grammar above, checking the Hoare-triple $\{\psi\}$ $S$ $\{\phi\}$ can be reduced to (decidable) logical formulas in the Satisfiability-Modulo-Theories (SMT) using standard techniques in program verification (Leino, 2010). One can use a predicate transformer called *weakest precondition WP* to convert the Hoare-triple to a decidable SMT formula that can be checked by Z3.

$$
\frac{\psi \implies WP(S, \phi)}{\{\psi\} \, S \, \{\phi\}}
$$

The *WP* is defined inductively on the structure of statements as follows:

$$
\begin{array}{rcl}
WP(x := a, \phi) & \doteq & \phi[a/x] \\
WP(\textbf{skip}, \phi) & \doteq & \phi \\
WP(S_1; S_2, \phi) & \doteq & WP(S_1, WP(S_2, \phi)) \\
WP(\textbf{if } b \textbf{ then } S_1 \textbf{ else } S_2, \phi) & \doteq & \bigwedge \begin{array}{l}(b \implies WP(S_1, \phi)) \\ (\neg b \implies WP(S_2, \phi))\end{array}
\end{array}
$$

### 3.2 Motivation and Problem Formulations

Given a problem definition $p$ that consists of preconditions $pre$, a loop **while** $b$ **do** $S$, and postconditions $post$, we can query LLMs to generate an invariant $i$ that satisfies the conditions specified in Equation (1). Although we have observed that

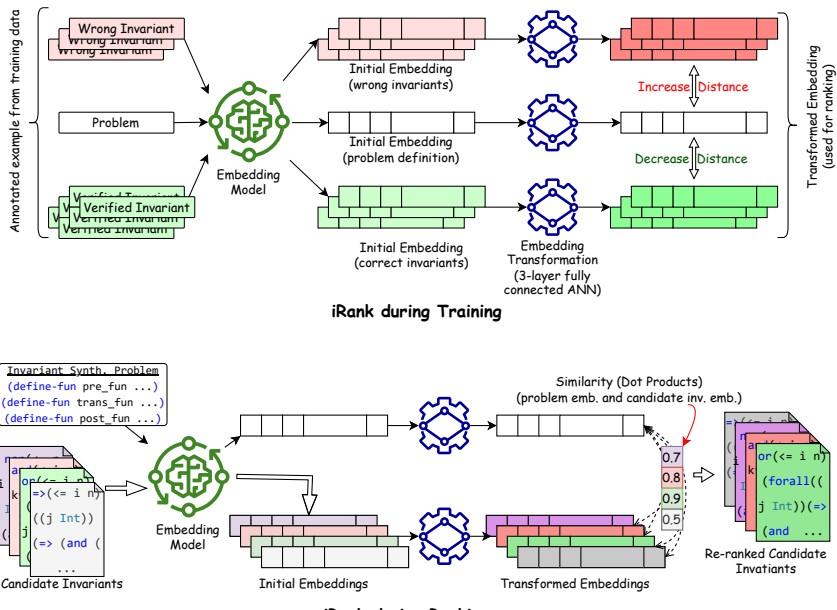

**Figure 1:** LLM for Loop Invariant synthesis.

LLMs are capable of producing loop invariants without syntax errors, they often require numerous samples before generating a correct invariant (we refer to Appendix B for a details). This results in inefficient resource utilization during the verification process, particularly when dealing with complex problem instances. More importantly, for a more practical scenario where generated invariants are used as part of an interactive verification system such as Dafny/F*, an incorrect invariant would take up valuable user time to perform a manual failed verification effort. Consequently, we propose the utilization of *iRank* to prioritize the generated invariants based on their likelihood of being correct. Figure 1 provides a high-level overview of the envisioned invariant generation-ranking system.

## 4  *iRank*: Methodology

The main intuition behind *iRank* is learning to pull the likely correct invariants to the front of a ranked list. Figure 1 shows a high-level overview of the ranker. We rely on a dataset, $\mathcal{D} = \{(p, I^+, I^-)\}$, containing Loop Invariant generation problem, $p$, a set of verified loop invariants, $I^+ = \{i^+ \mid i^+ \vdash p\}$, and a set of wrong loop invariant, $I^- = \{i^- \mid i^- \nvdash p\}$ for each of the problems, to build *iRank*. Our goal is to learn a function between a problem definition $p$ and invariant $i$, *i.e.,* $\sigma(p, i)$, which should satisfy the following constraint $\forall_{\{i^+, i^-\}} (\sigma(p, i^+) > \sigma(p, i^-))$.

**Contrastive Ranking.** To learn $\sigma$, we first ex-

tract the embedding of problem definitions and the invariants with an embedder, $\Phi$, *i.e.,* $x = \Phi(p)$, and $y = \Phi(i)$, where $x$ and $y$ are the embeddings of problem definition $p$, and invariant $i$, respectively. We learn a transformation function, $\Psi(x|\theta)$, which applies non-linear transformation on input vector $x$ with learnable parameter $\theta$. We then transform problem embedding $x$ to $x' = \Psi(x|\theta)$, and transform invariant embedding $y$ to $y' = \Psi(y|\theta)$. Now our target is to *maximize* the similarity between $x'$ and $y'$, when $y'$ corresponds to a correct invariant, *minimize* the similarity otherwise. We use the absolute cosine similarity as the measurement. Use of such allows us to set the maximum similarity to 1 (in the case of correct invariant) and the minimum to 0 (in the case of wrong invariant). We optimize the mean squared error loss to learn the parameters in $\Psi$. We experimented with two different embedding models based on LLMs, *i.e.,* text-embedding-ada-002 and davinci-similarity. Appendix A presents further details of *iRank*'s working procedure.

## 5  Experimental Design and Results

### 5.1  Experimental Setup

**Benchmarks.** We use Loop Invariant Synthesis benchmarks assimilated by Padhi et al. (2016) [1] constituting a set 940 challenge problems in Sy-

---

[1] https://github.com/SaswatPadhi/LoopInvGen/tree/master/benchmarks

Gus (Alur et al., 2013) format, with a SMT formula for the pre-condition, post-condition, and the transfer-function for the loop. We chose a SMT representation for our problem description $p$ to be agnostic to different encoding of C programs into logic. Among these problems, 541 were in the scope of LLM due to the context window size. We set the maximum context size as 4096 (with 3584 for prompt, 512 for generation).

**Gathering LLM-generated Invariants.** We conducted experiments involving two distinct language models: gpt-3.5-turbo and gpt-4. Our objective was to assess the capabilities of these language models out-of-the-box, and thus we employed a zero-shot prompting approach. This involved providing a problem description and an appropriate task explanation as a prompt (refer to Appendix C for an example). For each problem, we allowed both the models to generate invariants for a maximum duration of 10 minutes or until a verified invariant was found, whichever occurred first, resulting in solving 250 problems by gpt-3.5-turbo, and 188 problems for gpt-4[2]. It is important to clarify that the purpose of this paper is not to conduct a comparative analysis of these language models in relation to this specific problem. Instead, our objective is to propose a method to orthogonally augment LLM capabilities by reranking LLM generated invariants.

**Training Data.** We create the training dataset for *iRank* ($\mathcal{D} = \{(p, I^+, I^-)\}$) by combining invariants generated from different sources, such as different generations from LLMs, and invariants generated by LoopInvGen (Padhi et al., 2017). We divided the problems into five folds and trained 5 different rankers, one for each fold. During the evaluation, we select and load the trained model based on the problem under evaluation. Detailed statistics of data is available in Appendix A.3.

**Evaluation Metric.** We then sequentially attempt to check invariants from a ranked list. We evaluate three metrics – (i) $i^+$ ranks - rank of the correct invariant in the list, (ii) V@K - the percentage of problems where the verified invariant is found in top K invariants from the re-ranked list, and (iii) Number of Z3 calls - the total number of z3 calls before finding and reporting a correct invariant, a higher number of z3 calls indicate a high waste of computational resources.

---

[2]Note that the rate limit for gpt-4 was an order lower than gpt-3.5 in our usage resulting in an order less samples.

| Experiment | | $i^+$ ranks | | V@K (%) | | |
|---|---|---|---|---|---|---|
| | | Mean | Median | K=1 | K=5 | k=10 |
| LLM-ranks | | 189.78 | 62.00 | 5.2 | 11.6 | 18.4 |
| Expected ranks | | 95.35 | 31.02 | 8.0 | 19.2 | 25.2 |
| TF-IDF | | 103.45 | 24.00 | 17.6 | 32.0 | 38.8 |
| Emb. | Ada | 115.89 | 31.50 | 11.2 | 21.6 | 30.0 |
| | Davinci | 120.02 | 32.00 | 10.4 | 20.8 | 33.6 |
| *iRank* | Ada | 38.78 | 5.00 | 28.0 | 51.2 | 60.8 |
| | Davinci | **34.48** | **4.00** | **29.2** | **52.8** | **62.8** |

(a) Invariants generated by gpt-3.5-turbo .

| Experiment | | $i^+$ ranks | | V@K (%) | | |
|---|---|---|---|---|---|---|
| | | Mean | Median | K=1 | K=5 | k=10 |
| LLM-ranks | | 39.20 | 9.00 | 17.6 | 40.4 | 51.6 |
| Expected ranks | | 20.23 | 4.96 | 31.9 | 52.1 | 65.4 |
| TF-IDF | | 24.16 | 3.00 | 32.00 | 45.6 | 53.6 |
| Emb. | Ada | 20.69 | 5.50 | 26.6 | 51.1 | 64.9 |
| | Davinci | 23.56 | 5.00 | 27.7 | 52.1 | 63.3 |
| *iRank* | Ada | 13.18 | **2.00** | **44.7** | **74.4** | 81.4 |
| | Davinci | **11.96** | **2.00** | **44.7** | 71.8 | **81.9** |

(b) Invariants generated by gpt-4 .

**Table 1:** Comparison between different ranking strategies for re-ranking the invariants generated by different LLMs.

**Baselines.** *(a) LLM-ranks.* We take the invariants, in the order generated by the LLMs, as a ranklist. *(b) Expected-ranks.* We estimate the expected values of the evaluated metrics in this paper by randomly permuting the LLM-generated list of invariants (see Appendix D for more details). *(c) Embeddings.* We use the raw embeddings from LLM-based embedding models to calculate similarity without training. *(d) TF-IDF.* We use the textual similarity between the problem description and the candidate invariants for ranking.

**Research Questions.** In this paper, we studied two research questions. (i) How effective are LLM-based embeddings for ranking invariants? and (ii) Can a trained *iRank* help reduce the verification cost?

### 5.2 Results

Table 1 shows the quantitative evaluation of *iRank*. If we consider LLM-generated list of invariants as is, we observe that LLMs are able to generate a verified invariant after a significant number of wasted trials. For example, on average, gpt-3.5-turbo found an invariant after ∼190 failed attempt at generation. gpt-4 does much better, in comparison, with the mean rank of verified invariants being 39.20. The expected rank of the verified invariant from LLM-generations is 95.35 and 20.23, for gpt-3.5-turbo and gpt-4, respectively. The use of LLM-based embeddings (without any train-

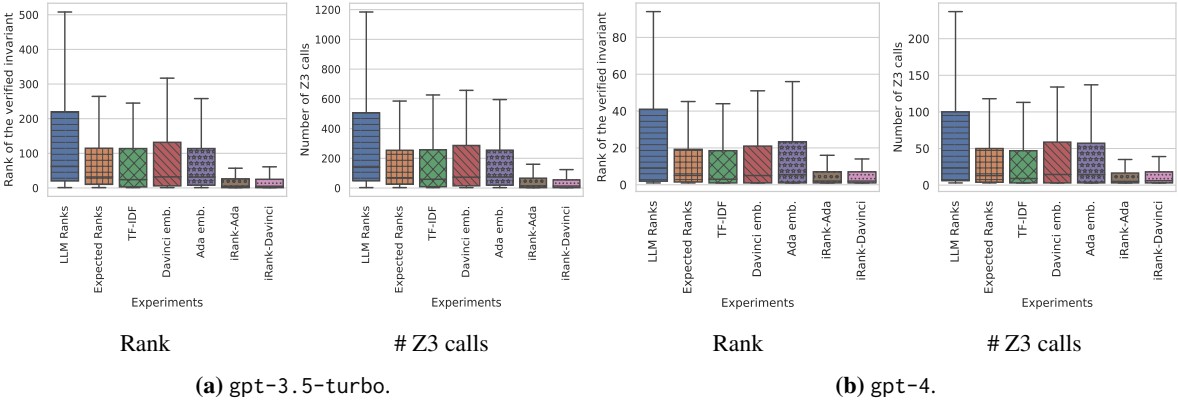

**(a)** `gpt-3.5-turbo`.                    **(b)** `gpt-4`.

**Figure 2:** Detailed results comparing ranks of the correct invariants and number of z3 calls.

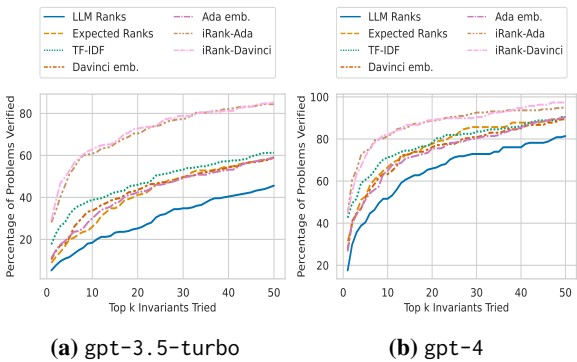

**(a)** `gpt-3.5-turbo`          **(b)** `gpt-4`

**Figure 3:** Percentage of problems solved w.r.t. number of invariants from the (re-)ranked list.

ing) such as from `text-embedding-ada-002` or `davinci-similarity` results is the mean rank of the verified invariant to be 115.89 and 120.02, respectively for `gpt-3.5-turbo`, and 20.69 and 23.56, respectively for `gpt-4`. While such a result looks like a significant improvement over LLM-ranks, it is slightly worse than expected ranks.

> ✓ LLM-based embeddings standalone may not serve well for reranking verified invariants.

The training procedure in *iRank* significantly contributes to improving the ranking of verified invariants by transforming the embeddings. The median rank of the verified invariant is 32 when using the embedding from `davinci-similarity` embeddings. With the contrastive training in *iRank*, the median rank is brought down to 4, showing a significant improvement. Such a trend persists across different embedding models and generations from different LLMs. Figure 2a, and Figure 2b shows the detailed results invariants generated by `gpt-3.5-turbo` and `gpt-4`. In both trained and raw embeddings, the dif-

ference between `text-embedding-ada-002` and `davinci-similarity` models, the performance differences are not different with statistical significance (with p-value > 0.1). In both models' cases, there is no statistically significant difference between the expected rank and raw embedding-based ranking. Note that, similar to the existing works (Ryan et al., 2020), we use z3 call to measure resource wastage. However, depending on the system the experiments are carried on, the time saved from reranking with *iRank* could be different. We report our experimental results of wall clock time in Appendix B (Table 2).

Figure 3 shows the percentage of problems verified after trying k invariants (V@K). We observe that the *iRank* curves are very steep at the beginning of the curves compared to the baseline, signifying that it could rank the verified invariants in significantly higher positions than baselines.

> ✓ Contrastive training in *iRank* brings the verified invariant closer to the problem while pushing the wrong ones resulting in a significant reduction in the verification cost.

## 6 Conclusion

We presented a novel approach, *iRank*, to rank the loop invariants generated by LLMs based on their likelihood of being correct. Our ranker leverages a contrastive learning technique to discriminate between correct and incorrect invariants. Our evaluation demonstrates that our approach significantly reduces the invariant checking effort compared to the original LLM outputs.

## Limitations

**Assumptions of LLM inference cost.** In this paper, we assumed the cost of calling LLMs is negligible compared to the cost of calling the verifier for checking an invariant. Current access to LLMs (at least the one we studied in the paper) is available through the rest API, which can be scaled up by the API vendor with distributed processing of LLM. However, with the increase in the problem complexity, *i.e.,* non-linear invariants, high number of variables, the check for correctness of an invariant become exponentially more expensive. In the case where call to LLM is much more expensive than LLM, iRank will reduce the number of Z3 calls, but may not contribute to actual cost savings.

**Comparison with state-of-the-art (SOTA) invariant synthesis.** The goal of this paper is *not* to establish SOTA for loop invariant synthesis. In contrast, we investigate LLMs' capacity to generate Loop invariant relying on their emergent behavior. *iRank* is proposed as an orthogonal tool and evaluated to rank LLM generations in this paper. However, in theory, *iRank* should be able to rerank invariants generated by any generator. Nevertheless, the design of the SOTA technique of Loop Invariant Synthesis with LLM (perhaps with other tools) remain an open problem, which we leave for future research.

**Stability of LLM predictions.** Due to the stochastic (and nondeterministic [3]) nature of the LLM, especially in higher temp, we observe unstable generation from the LLM. Nevertheless, we evaluated the results from one sample run from `gpt-3.5-turbo` and one from `gpt-4` as case studies. While we acknowledge the possibility of unstable behavior, similarity in the performance trend (*i.e., iRank*'s performance improvement over LLM and expected ranks, also over raw embeddings) give us confidence about the impact of *iRank*.

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

| Experiment | | i⁺ ranks | | V@K (%) | | | Time(s) (Mean/Median) | | | |
|---|---|---|---|---|---|---|---|---|---|---|
| | | Mean | Median | K=1 | K=5 | k=10 | Embed | Ranking | Verification | Total |
| LLM-ranks | | 189.78 | 62.00 | 5.2 | 11.6 | 18.4 | 0 / 0 | 0 / 0 | 7.33 / 2.23 | 7.33 / 2.23 |
| Expected ranks | | 95.35 | 31.02 | 8.0 | 19.2 | 25.2 | 0 / 0 | 0 / 0 | 3.67 / 1.12 | 3.67 / 1.12 |
| TF-IDF | | 103.45 | 24.00 | 17.6 | 32.0 | 38.8 | 0 / 0 | 0.05 / 0.02 | 3.83 / 0.94 | 3.88 / 1.00 |
| Emb. | Ada | 115.89 | 31.50 | 11.2 | 21.6 | 30.0 | 1.29 / 0.42 | 0.05 / 0.01 | 4.43 / 1.23 | 5.78 / 1.82 |
| | Davinci | 120.02 | 32.00 | 10.4 | 20.8 | 33.6 | 9.17 / 3.02 | 0.45 / 0.15 | 4.79 / 1.20 | 14.41 / 4.48 |
| *iRank* | Ada | 38.78 | 5.00 | 28.0 | 51.2 | 60.8 | 1.29 / 0.42 | 0.06 / 0.02 | 1.64 / 0.19 | **2.98 / 0.97** |
| | Davinci | **34.48** | **4.00** | **29.2** | **52.8** | **62.8** | 9.17 / 3.02 | 0.48 / 0.15 | 1.28 / 0.16 | 10.93 / 3.68 |

**(a)** Invariants generated by `gpt-3.5-turbo` .

| Experiment | | i⁺ ranks | | V@K (%) | | | Time(s) (Mean/Median) | | | |
|---|---|---|---|---|---|---|---|---|---|---|
| | | Mean | Median | K=1 | K=5 | k=10 | Embed | Ranking | Verification | Total |
| LLM-ranks | | 39.20 | 9.00 | 17.6 | 40.4 | 51.6 | 0 / 0 | 0 / 0 | 1.61 / 0.24 | 1.61 / 0.24 |
| Expected ranks | | 20.23 | 4.96 | 31.9 | 52.1 | 65.4 | 0 / 0 | 0 / 0 | 0.83 / 0.19 | 0.83 / 0.19 |
| TF-IDF | | 24.16 | 3.00 | 32.00 | 45.6 | 53.6 | 0 / 0 | 0.01 / 0.006 | 0.80 / 0.11 | 0.81 / 0.12 |
| Emb. | Ada | 20.69 | 5.50 | 26.6 | 51.1 | 64.9 | 0.23 / 0.06 | 0.01 / 0.004 | 0.67 / 0.19 | 0.94 / 0.26 |
| | Davinci | 23.56 | 5.00 | 27.7 | 52.1 | 63.3 | 1.93 / 0.48 | 0.12 / 0.03 | 0.75 / 0.16 | 2.80 / 0.77 |
| *iRank* | Ada | 13.18 | **2.00** | **44.7** | **74.4** | 81.4 | 0.25 / 0.06 | 0.02 / 0.004 | 0.44 / 0.06 | **0.71 / 0.16** |
| | Davinci | **11.96** | **2.00** | **44.7** | 71.8 | **81.9** | 1.93 / 0.48 | 0.13 / 0.03 | 0.74 / 0.06 | 2.80 / 0.72 |

**(b)** Invariants generated by `gpt-4` .

**Table 2:** Comparison between different ranking strategies for re-ranking the invariants generated by different LLMs.

Xujie Si, Hanjun Dai, Mukund Raghothaman, Mayur Naik, and Le Song. 2018. Learning loop invariants for program verification. *Advances in Neural Information Processing Systems*, 31.

Nikhil Swamy, Juan Chen, Cédric Fournet, Pierre-Yves Strub, Karthikeyan Bhargavan, and Jean Yang. 2011. Secure distributed programming with value-dependent types. In *Proceedings of the 16th ACM SIGPLAN International Conference on Functional Programming*, ICFP '11, page 266–278, New York, NY, USA. Association for Computing Machinery.

Jianan Yao, Gabriel Ryan, Justin Wong, Suman Jana, and Ronghui Gu. 2020. Learning nonlinear loop invariants with gated continuous logic networks. In *Proceedings of the 41st ACM SIGPLAN Conference on Programming Language Design and Implementation*, pages 106–120.

| Experiment | | i⁺ ranks | | V@K (%) | | |
|---|---|---|---|---|---|---|
| | | Mean | Median | K=1 | K=5 | k=10 |
| Expected ranks | Original | 95.35 | 31.02 | 8.0 | 19.2 | 25.2 |
| | **Deduplicated** | **65.24** | **24.07** | **8.4** | **22.8** | **31.2** |
| *iRank*-ada | Original | 38.78 | 5.00 | 28.0 | 51.2 | 60.8 |
| | **Deduplicated** | **18.79** | **4.00** | **28.4** | **56.0** | **65.6** |

**(a)** Invariants generated by `gpt-3.5-turbo` .

| Experiment | | i⁺ ranks | | V@K (%) | | |
|---|---|---|---|---|---|---|
| | | Mean | Median | K=1 | K=5 | k=10 |
| Expected ranks | Original | 20.23 | 4.96 | 31.9 | 52.1 | 65.4 |
| | **Deduplicated** | **13.99** | **4.89** | **31.4** | **53.7** | **72.8** |
| *iRank*-ada | Original | 13.18 | 2.00 | 44.7 | 74.4 | 81.4 |
| | **Deduplicated** | **8.73** | **2.00** | **46.8** | **77.1** | **86.7** |

**(b)** Invariants generated by `gpt-4-turbo` .

**Table 3:** Ranking result of correct invariant by de-duplicating semantic equivalent candidates

# A Further Details of *iRank*

In this section, we present a comprehensive overview of the operational workflow of *iRank*, as visualized in Figure 1.

## A.1 Training *iRank*

As elucidated in Section 4, the training of *iRank* necessitates a dataset containing invariant synthesis problems, akin to those illustrated in Figure 4. Each problem in the training dataset requires at least one correct invariant and a set of incorrect invariants, all expressed in the SyGus format (refer to Figure 5 for an example). We employ the specified embedding model, namely `text-embedding-ada-002` or `davinci-similarity`, to acquire initial embeddings for both the problems and candidate solutions. These initial embeddings undergo transformation via a three-layered fully connected feedforward network to yield transformed embeddings. The training objective is twofold: minimize the distance between the transformed embedding of the problem and the corresponding correct solutions, while maximizing the distance from incorrect ones. Once this model is trained, it is employed to rank the candidate invariants generated by the LLM or any other generator.

## A.2 Ranking with *iRank*

Upon successful training of the transformation network within *iRank*, it is used for ranking pur-

| Problem Statistics | |
| --- | --- |
| Total Problems | 541 |
| **Problem Types Statistics** | |
| Linear Integer (LIA) | 496 (91.68%) |
| Non-Linear Integer (NIA) | 27 (4.99%) |
| Array Linear Integer (ALIA) | 18 (3.33%) |
| **Problem Semantics Statistics** | |
| Number of functions | Min = 3, Max = 9 |
| Number of Variables | Min = 2, Max = 90 |
| Variable types | Integer = 80.31%
Boolean = 19.31%
Array[Integer] = 0.36%
Array[Boolean] = 0.03% |
| **Operator Statistics (of the correct invariants)** | |
| Conjuctions | 43% (of all operators)
1.12 (avg. per example) |
| Disjunctions | 20% (of all operators)
0.53 (avg. per example) |
| Negations | 37% (of all operators)
0.96 (avg. per example) |
| Addition/Subraction | 43% (of all operators)
0.5 (avg. per example) |
| Multiplication/Division | 3.1% (of all operators)
0.18 (avg. per example) |
| Logical Comparison | 88.5% (of all operators)
5.25 (avg. per example) |
| **Length Statistics** | |
| Problem Length | Min = 92 (N), 88 (T)
Max = 2732 (N), 3146 (T)
Mean = 740 (N), 922 (T) |
| Invariant Length (gpt-3.5-turbo) | Min = 18 (N), 15 (T)
Max = 605 (N), 506 (T)
Mean = 110 (N), 125 (T) |
| Invariant Length (gpt-4) | Min = 18 (N), 15 (T)
Max = 513 (N), 488 (T)
Mean = 90 (N), 102 (T) |

N = Tokenized with NLTK
T = Tokenized with TikToken

**Table 4:** Stattistics of the experimental data

| | |
| --- | --- |
| Number of Layers | 3 |
| Hidden Size | 1536 (`text-embedding-ada-002`)
12288 (`davinci-similarity`) |
| Optimizer | Adam |
| # Training Epochs | 20 |
| Weight Decay | 0.001 |
| Max Gradient Norm | 1.0 |
| Learning Rate | $5 * 10^5$ |
| LR Scheduler Type | Linear |
| Warmup Steps | 500 |

**Table 5:** Hyperparameter for Model and Training

poses. To rank the candidate invariants, *iRank* initially extracts the initial embedding of the problem and a list of solutions, using the same embedding model (`text-embedding-ada-002` or `davinci-similarity`) as employed during training. The trained transformation network is then used to transform these embeddings. These transformed embeddings serve as vectors for the re-ranking process, where *iRank* calculates the cosine similarity between the transformed embedding of the problem and each of the candidate solutions. The candidate solutions are then sorted and returned based on their similarity with the problem.

### A.3 Data Details and Training Hyperparameters

Table 4 shows the statistics of the data we used to experimentally evaluate *iRank*. Table 5 shows the hyperparameters for the models and training we

used in this study.

## B  Detailed Results

In addition to comparing the number of Z3 calls, we compared the wall clock time. Table 2 shows a comparison of time (as an extension of Table 1). We conducted all the experiments in 24 cores AMD Epyc 7V13 Linux server running on Linux Ubuntu 20.04 LTS with 220 GB RAM, and a single NVIDIA A100-PCIE-80GB GPU. For LLM-Ranks, Expected ranks there is no embedding and ranking, thus the verification time is the bottleneck. For TF-IDF, while there is no embedding time, the is a little bit of ranking time.

The Ada embedding time in *iRank* was very small compared to the Davinci embedding, thus, in the case of *iRank*-ada, embedding and ranking time was offset by the time *iRank* reduced in the verification effort. In contrast, the Davinci embedding in *iRank* is more expensive than the reduction in the verification time, resulting in a worse wall clock time performance than the LLM ranks. We conjecture that the `text-embedding-ada-002` (embedding dim = 1536) is a smaller model compared to `davinci-similarity` (embedding dim = 12188), thus requiring significantly longer time to embed an input sequence (problem description or invariant).

It is important to note here that, this experiment is *only* meant for an illustration of potential threats to *iRank*, and is dependent on a lot of variables, including, but not limited to OpenAI subscription maximum rate limit, network latency for initial embeddings, etc.

In addition, we analyzed the generated invariant candidates from LLMs, and removed any semantic duplicates. Given two invariant candidates $i_a$ and $i_b$ parameterized by set of variables $\{v_1, \ldots v_n\}$,

we define semantic equivalence as,

$$\forall v_1, \ldots, v_n : i_a(v_1, \ldots, v_n) \Leftrightarrow i_b(v_1, \ldots, v_n)$$

For comparing equivalence of two candidate invariants, we make one call to the z3. Such a semantic deduplication requires comparison of a newly generated candidate invariant with all previous candidates, necessitating $\Theta(n^2)$ z3 calls, just to deduplicate. Table 3 shows the results on deduplicated candidates in comparison with the original list of invariants. As expected, after deduplicating, the expected ranks improves. Interestingly, even in the list of candidates where all candidates are semantically unique, *iRank* improves the rank of the correct invariant, resulting in higher $V@K$.

## C   Illustrative example

As an illustration of our proposed approach, we present an example from FiB (Lin et al., 2017) benchmark [4]. The problem is represented in SyGus format as shown in Figure 4.

```
(set-logic LIA)

(synth-inv inv_fun ((i Int) (n Int) (a Int) (b Int)))

(define-fun pre_fun ((i Int) (n Int) (a Int) (b Int)) Bool
    (and
        (= i 0) (= a 0) (= b 0) (>= n 0)
    )
)
(define-fun trans_fun ((i Int) (n Int) (a Int) (b Int)
        (i! Int) (n! Int) (a! Int) (b! Int)) Bool
    (or
        (and
            (< i n) (= i! (+ i 1)) (= a! (+ a 1))
            (= b! (+ b 2)) (= n! n)
        )
        (and
            (< i n) (= i! (+ i 1)) (= a! (+ a 2))
            (= b! (+ b 1)) (= n! n)
        )
        (and
            (>= i n) (= i! i) (= a! a) (= b! b) (= n! n)
        )
    )
)
(define-fun post_fun ((i Int) (n Int) (a Int) (b Int)) Bool
    (=> (not (< i n)) (= (+ a b) (+ (+ n n) n))))

(inv-constraint inv_fun pre_fun trans_fun post_fun)

(check-synth)
```

**Figure 4:** Invariant synthesis problem in FiB-8.sl

We create the following prompt to call LLMs.

```
Here is a loop invariant synthesis problem
in SyGus format.

<<<<The problem definition from above>>>>>
```

[4] https://github.com/spencerxiao/
ase2017-results-and-tools/tree/master/FiB_Tool/
benchmarks

```
Synthesize a necessary and sufficient invariant.

Start the invariant with
"(define-fun inv_fun ((i Int) (n Int) (a Int)
(b Int)) Bool (" and end with ")".

Surround only the invariant with  and
. You don't need to explain the invariant,
just synthesize it.
```

The `gpt-3.5-turbo` model generated invariant shown in Figure 5 after 144 unsuccessful attempts.

```
(define-fun inv_fun ((i Int) (n Int) (a Int) (b Int))
Bool
    (or
        (and (<= i n) (= (+ a b) (* 3 i)))
        (and (> i n) (= (+ a b) (* 3 n)))
    )
)
```

**Figure 5:** Correct invariant generated by `gpt-3.5-turbo`.

The `gpt-4` model generated the invariant shown in Figure 6 after 2 unsuccessful attempts.

```
(define-fun inv_fun ((i Int) (n Int) (a Int) (b Int))
Bool
    (and (<= i n) (= (+ a b) (* i 3)))
)
```

**Figure 6:** Correct invariant generated by gpt-4.

*iRank* trained based on `text-embedding-ada-002` repositioned the `gpt-3.5-turbo` at 8th position in the list and the `gpt-4` generated correct invariant in position 2. Note that we show this example only for illustration purposes.

## D   Experiment on Expected re-ranking

The list of invariants generated by LLM as a ranked list could be unstable and susceptible to variations in performance across different experiments. Thus, as described in Section 5.1, we estimate the expected ranks by randomly permuting the list. Ideally, to get a perfect estimation, we should consider all possible permutations of the list, which can be very expensive (exponential order on the number of elements in the list). Figure 7 shows ablation of mean rank *w.r.t.* the number of random permutations. As we can see, with a gradual increase in the number of permutations, the variance in the metrics gradually reduces, *i.e.,* the metrics converges. Throughout the paper, we set the number of permutations to be 100 for estimating the expected rank metrics.

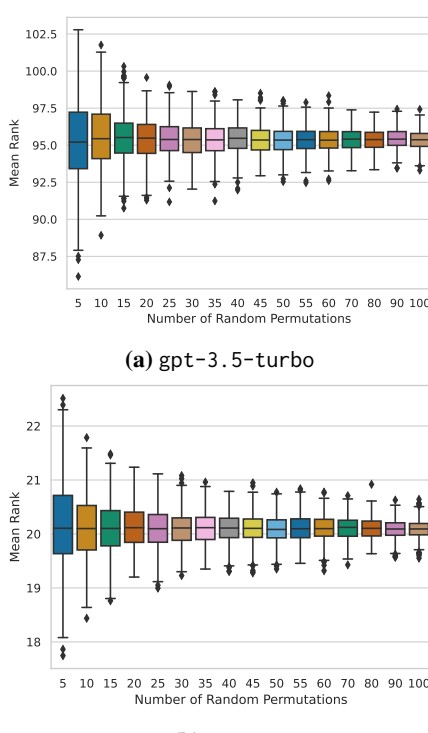

**(a)** `gpt-3.5-turbo`

**(b)** `gpt-4`

**Figure 7:** Result stabilization with an increasing number of random permutations

## E    Visualization of the impact of training in *iRank*

Figure 8 show a t-SNE plot of the raw LLM embeddings and the transformed embeddings for a few example problems. For the first three examples(Figures 8a, 8b, 8c, respectively), *iRank* brings the correct invariant closer to the problem than any other invariants. For the example presented in Figure 8d, *iRank* could not make the correct invariant as the closest to the problem. While there are cases where *iRank*'s transformation fails to bring the correct invariant in the closest proximity, in most cases, it can bring correct invariants closer to the transformed problem embedding, as corroborated by the results in Appendix B

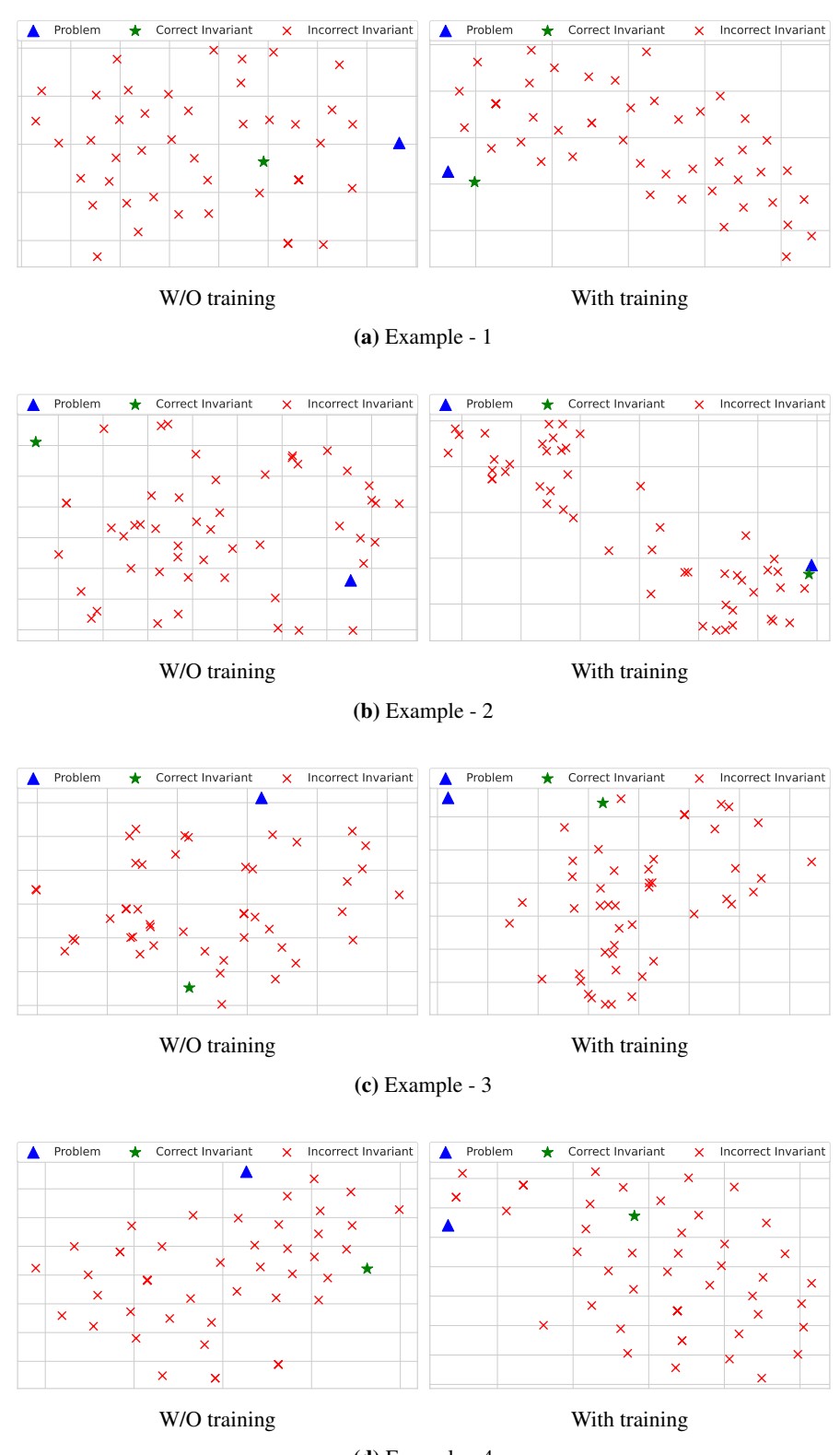

**Figure 8:** t-SNE plots of embeddings with and without training for a few example problems. The number of incorrect invariants is downsampled for better visualization clarity.

