# OpenReview forum: "Ranking LLM-Generated Loop Invariants for Program Verification"
_EMNLP/2023/Conference — EMNLP 2023 Findings_

### Official Review · Reviewer_p5MA · 2023-08-02

**Soundness:** 3

**Excitement:**

3: Ambivalent: It has merits (e.g., it reports state-of-the-art results, the idea is nice), but there are key weaknesses (e.g., it describes incremental work), and it can significantly benefit from another round of revision. However, I won't object to accepting it if my co-reviewers champion it.

**Missing References:**

References are great!

**Paper Topic And Main Contributions:**

The paper is about improving a LLM's ability to generate loop invariants for programs. The authors:
1) use existing loop invariant synthesis benchmarks to curate a training set of invariants
2) use contrastive learning to train a ranking model to compare various generated invariants
3) show that the ranker can improve the ability to choose the correct invariant by measuring a) the mean rank of the correct invariant among a list, and b) the percentage of problems where the invariant is found among the top-K invariants.

**Questions For The Authors:**

A - Could you give an analysis of wall-clock time? For example, what is the average time needed to find the correct invariant (including LLM generation time, ranking time, checking time) in the Emb. and iRank scenarios in Table 1?

B - Could you give more details on how iRank was trained? What is the architecture on top of the un-trained embeddings? How many epochs was it trained over? What was the optimizer? How are the data folds different, and how much data was there?

C - There has been a long line of work prior to yours from the PL literature on synthesizing loop invariants. How do you envision combining your method with some of these existing techniques?

**Reasons To Accept:**

1. To my knowledge, this is the first work that tackles the problem of generating loop invariants in the SMT format automatically with LLMs. The closest to this is a previous work on generating loop invariants (Pei et al) and only considers Java.
2. The authors introduce a training and evaluation set for this task that would allow for future advancements in this direction.
3. The authors propose an interesting contrastive learning training framework that pulls correct invariants closer to the embedding of the problem and incorrect invariants further. It is simple and intuitive.
4. The evaluation is sound and shows that the framework in 3) improves the ability to identify correct invariants.

**Reasons To Reject:**

1. One of the main claims made by the paper is that iRank can lead to a significant reduction in the verification cost. The authors cite the number of Z3 calls as evidence of this, but there are additional dimensions of wall-clock time and monetary cost. Calling a LLM would certainly lead to increased monetary cost, and there is no analysis of wall-clock time reported.
2. There is no qualitative analysis, so it is hard to understand the effect of iRank in depth. For example, does the LLM generates many obviously incorrect solutions that even simple heuristics could filter out a majority of the incorrect ones? Perhaps iRank is very effective for a certain class of invariants but not for others, but there is no way to tell that from what is presented here.
3. The data details, architecture details, and training details of iRank are unclear.
4. From my perspective, one of the main contributions of the paper (aside from ranking) is the introduction of this task for SMT invariant synthesis. The authors do curate a dataset, but do not give any details analyzing this dataset (e.g. a taxonomy of problems that are included in the dataset, difficulty of problems, histogram of lengths, etc.)

**Reproducibility:**

4: Could mostly reproduce the results, but there may be some variation because of sample variance or minor variations in their interpretation of the protocol or method.

**Reviewer Confidence:**

4: Quite sure. I tried to check the important points carefully. It's unlikely, though conceivable, that I missed something that should affect my ratings.

**Typos Grammar Style And Presentation Improvements:**

It would be great if Figure 1 included the architecture details of the embedding model.

---

> ### Author Rebuttal · Authors · 2023-08-29
>
> We thank the reviewer for their insightful comments and suggestions. Here are the answers to the reviewer’s questions.
>
> 1. **Comparison of Time:** Following the reviewer’s suggestion, we performed an experiment, where the metric is time. In particular, we measure the verification time without the ranking (i.e., following the LLM-rank). When we do rank (either based on raw embeddings, or based on trained embedding), we calculate the time to embed, rank, and verification time until a correct invariant is found. Below are our analysis results.
>
>     Below are the results for ranking candidate invariants generated by `gpt-3.5-turbo`.
>
>     | Experiment | Mean $i^+$ rank | $V@1$ | $V@10$ | Embedding Time | Ranking Time | Verification Time | Total Time |
>     |------------------|----------------|-------|--------|----------------|--------------|-------------------|------------|
>     | LLM-Ranks  | $189.78$ | $5.2$ | $18.4$ | $0$ | $0$ | $7.33$ | $7.33$ |
>     | Expected Ranks | $95.33$ | $8.0$ | $25.2$ | $0$ | $0$ | $3.67$ | $3.67$ |
>     | Emb. Ada  | $115.89$ | $11.2$  	| $30.0$   	| $1.29$           	| $0.05$         	| $4.43$              	| $5.78$       	|
>     | Emb. Davinci | $120.02$          	| $10.4$  	| $33.6$   	| $9.17$           	| $0.45$         	| $4.79$              	| $14.41$      	|
>     | $iRank$-ada | $38.78$           	| $28.0$  	| $60.8$   	| $1.29$           	| $0.06$         	| $1.64$              	| $2.98$       	|
>     | $iRank$-davinci | $34.48$           	| $29.2$  	| $62.8$   	| $9.17$           	| $0.48$         	| $1.28$              	| $10.93$      	|
>
>     Below are the results for ranking candidate invariants generated by `gpt-4`.
>
>     | Experiment | Mean $i^+$ rank | $V@1$ | $V@10$ | Embedding Time | Ranking Time | Verification Time | Total Time |
>     |------------------|-----------------|-------|--------|----------------|--------------|-------------------|------------|
>     | LLM-Ranks | $39.20$           	| $17.6$  	| $51.6$   	| $0$              	| $0$            	| $1.61$              	| $1.61$       	|
>     | Expected Ranks | $20.23$           	| $31.9$  	| $65.4$   	| $0$              	| $0$            	| $0.83$              	| $0.83$       	|
>     | Emb. Ada | $20.69$           	| $26.6$  	| $64.9$   	| $0.23$           	| $0.01$         	| $0.67$              	| $0.94$       	|
>     | Emb. Davinci | $23.56$           	| $27.7$  	| $63.3$   	| $1.93$           	| $0.12$         	| $0.75$              	| $2.80$       	|
>     | $iRank$-ada | $13.18$           	| $44.7$  	| $81.4$   	| $0.23$           	| $0.02$         	| $0.44$              	| $0.72$       	|
>     | $iRank$-davinci | $11.96$           	| $44.7$  	| $81.9$   	| $1.93$           	| $0.13$         	| $0.74$              	| $2.80 $      	|
>
>     As suggested by the reviewer, we augmented the Table-1 with wall clock time. As we can see from the table, without any ranking (as generated by the `gpt-3.5`), the mean time to find a correct invariant if $7.33s$, all of which are accounted for the calls to Z3 for verifying correctness of candidate invariants. With iRank, there is an overhead of embedding and ranking. For $iRank-ada$, median time to find the correct invariant is $2.98$, significantly lower than the time needed from LLM-ranks. However, we observe that, for $iRank-davinci$, the time is more than that of LLM-ranks. Investigating further, we observe that while iRank-davinci does improve the verification time, embedding time for `davinci-similarity` is significantly higher. There could be several explanations behind such behavior. Unfortunately, since we used OpenAI API for getting the embedding, we can only conjecture the explanation. Nevertheless, we believe `davinci-similarity` (embedding dimension = $12288$) embedding model is a much bigger model than `text-embedding-ada-002` (embedding dimension = $1536$). In addition, network latency, rate limit of the OpenAI subscription plans could be issues. Due to these factors beyond our control, we report the number of z3 calls. In this paper, we show that while LLM can be used to generate invariant, a simple ranker could save a lot of wasted verification effort. We do hope this research will augment future research on generate and validate based invariant synthesis.
>
> 2. **Quality of the LLM-generated invariants:** To understand LLM-generated invariants qualitatively, we ought to investigate how semantically diverse invariants are generated by the LLMs. Identifying semantic diversity is a non-trivial task - to simplify the problem, we identify semantically duplicate invariants from the candidate invariants generated by the LLMs.  We evaluate whether two invariant $i_1$ and $i_2$ semantic equivalent by evaluating the following SMT formula with Z3, $\forall{v_1, v_2, …, v_n}(i_1(v_1, v_2, …, v_n) \Leftrightarrow i_2(v_1, v_2, …, v_n))$. We find that `gpt-3.5-turbo` generated $9.29$% semantically duplicate invariants, and `gpt-4` generated $4.70$% semantically duplicate invariant – showing that LLMs generate diverse set of invariant candidates.
>
> 3. **Details about the model and training:** We apologize for the missing details. We have uploaded the redrawn image articulating more details about the $iRank$ architecture and training/ranking method [in this anonymous link](https://figshare.com/ndownloader/files/42163257/preview/42163257/preview.jpg). For the embedding transformation, we use a 3-layered fully connected feedforward ANN. The hidden dimension of the embedding transformation model depends on the initial embedding model. For instance, when we use embeddings from `text-embedding-ada-002` model, the hidden dimension is $1536$. Also note that, since we are extracting initial embeddings through OpenAI API, the only learnable parameters in iRank are in the Embedding Transformation module. After we extract the initial embeddings for both the problem definition and the invariant candidates from OpenAI embedding models using their API, we transform them using the $iRank$ embedding transformation module . We optimize $iRank$ to bring the transformed embedding of the problem closer to the correct invariant and further away from the incorrect invariant from the training set. To optimize we use the `MSELoss` where the expected similarity of the problem and correct invariant is 1 and with incorrect invariants, it is 0. Here are the training details.
>     ```json
>     {
>         "optimizer": "Adam",
>         "learning_rate": 5e-05,
>         "num_train_epochs": 20,
>         "weight_decay": 0.001,
>         "max_grad_norm": 1.0,
>         "lr_scheduler_type": "linear",
>         "warmup_steps": 500,
>         ...
>     }
>     ```
>     We pass the training parameters as a user input and pass these parameters as a json configuration to the training. We release the default configuration (including these values mentioned above) as part of our artifact release.  We evaluate iRank based on a 5-fold cross validation. In each of these 5 experiments, the set of problems used in training and in evaluation are completely separate.
>
> 4. **Details about the dataset:**
> We evaluate iRank on the SyGus benchmark, which includes the loop invariant synthesis problem from several different sources, including different verification challenges. The problems include _Linear Integer Arithmetic_ (91.68%, 496 out of 541), _Array Linear Integer Arithmetic_ (3.33%, 18 out of 541) and _Non-linear Integer Arithmetic_ (4.99%, 27 out of 541) problems. Here are more detailed statistics of the problems in the dataset.
>
>     | Problem Type Statistics         |                |
>     |---------------------------------|----------------|
>     | Linear Integer Arithmetic       | $496$ ($91.68$%) |
>     | Non-Linear Intehger Arithmetic  | $29$ ($4.99$%)   |
>     | Array Linear Integer Arithmetic | $18$ ($3.33$%)   |
>
>     | Problem Semantics Statistics |                          |
>     |------------------------------|--------------------------|
>     | Number of functions          | Min = $3$, Max = $9$     |
>     | Number of variables          | Min = $2$, Max = $90$    |
>     | Variable Types               | Integer = $80.31$%       |
>     |                              | Boolean = $19.31$%       |
>     |                              | Array[Integer] = $0.36$% |
>     |                              | Array[Boolean] = $0.03$% |
>
>     | Operator Statistics (of correct invariants)        | % of all operators | Avg. # per examples |
>     |----------------------------------------------------|--------------------|--------------------------|
>     | Conjuctions                                        | $43$                 | $1.12$                     |
>     | Disjunctions                                       | $20$                 | $0.53$                    |
>     | Negations                                          | $37$                 | $0.96$                     |
>     | Addition/Subraction                                | $43$                 | $0.5$                      |
>     | Multiplication/Division                            | $3.1$                | $0.18$                     |
>     | Logical Comparison                                 | $88.5$               | $2.25$                     |
>     | Examples with quantifiers ($\forall$ or $\exists$) | $7$                  |
>
>     | Length Statistics           | Min | Max  | Avg |
>     |-----------------------------|-----|------|-----|
>     | Problem Length (NLTK)       | $92$  | $2732$ | $740$ |
>     | Problem Length (TokToken)   | $88$  | $3146$ | $922$ |
>     | Invariant Length (NLTK)     | $18$  | $605$  | $110$ |
>     | Invariant Length (TikToken) | $15$  | $506$  | $125$ |
>
>
> 5. **Integration of iRank with Symbolic technique and Future direction:** There is long line of work with symbolic heuristics for synthesizing Loop Invariants falling into mostly two broad categories (a) generate and validate (G&V) based, and (b) abstract interpretation based. We envision our technique augmenting the generate and validate line of work in invariant synthesis. iRank could be an orthogonal addition to the generation in the G&V approach, where after generation, iRank reorders for effectively reducing the cost of validation.
>
> Once again, we thank the reviewer for the thoughtful comments and reviews. We hope our answers clarifies the reviewer's questions. We will incorporate these changes in te camera ready version.

---

### Official Review · Reviewer_P8En · 2023-08-03

**Soundness:** 3

**Excitement:**

3: Ambivalent: It has merits (e.g., it reports state-of-the-art results, the idea is nice), but there are key weaknesses (e.g., it describes incremental work), and it can significantly benefit from another round of revision. However, I won't object to accepting it if my co-reviewers champion it.

**Paper Topic And Main Contributions:**

This paper proposes a re-ranker model for LLMs, with the goal of distinguishing between correct inductive invariants and incorrect attempts based on the problem definition. The ranker is optimized using contrastive loss. The authors conduct various experimental results showing that their model leads to a reduction in the number of calls to a verifier.

**Questions For The Authors:**

See weaknesses section above.

**Reasons To Accept:**

- The literature review seems comprehensive to identify prior work in synthesizing loop invariants for program verification, as well as providing a background in recent LLM models. Authors also motivate their problem setting by identifying limitations with the existing work and with the Background & Motivation section.

**Reasons To Reject:**

- It would help if the authors could more concretely list out their contributions as a section/subsection so it would be easier to follow.
- Figure 1b for iRank’s architecture seems somewhat abstract and not that help to follow for understanding the architecture. Can the authors include more model details in this figure?
- Further, the model details are too ambiguous to be able to reproduce the paper. For example, it is simply mentioned that an embedding model is used to extract the embedding of problem definitions and the invariants, but no details provided about the embedding model. Furthermore, it is just mentioned that a Transformation function is applied but no details regarding it. As a result, it is also difficult to determine the contribution and novelty of the proposed method with these important elements missing.
- The datasets seem to be very small in the range of hundreds or thousand questions. What is the scalability of the proposed model? Can the authors provide some complexity analysis?

**Reproducibility:**

3: Could reproduce the results with some difficulty. The settings of parameters are underspecified or subjectively determined; the training/evaluation data are not widely available.

**Reviewer Confidence:**

3: Pretty sure, but there's a chance I missed something. Although I have a good feel for this area in general, I did not carefully check the paper's details, e.g., the math, experimental design, or novelty.

---

> ### Author Rebuttal · Authors · 2023-08-29
>
> We thank the reviewer for their insightful comments and suggestions. Here are the answers to the reviewer’s questions.
>
> 1. **Contribution of the paper:** The main contributions in the paper are,
>     1. In this paper we studied the feasibility of LLMs such as (`gpt-3.5-turbo`, and `gpt-4`) for generating loop invariants in SMT-2 (SyGus) format in a language independent manner. Our study shows that with simple prompts LLMs are able to generate correct invariant in upto 46% of the cases ($250$ out of $541$ problems).
>     2. Based on our observation that the correct invariant is generated after lots of failed attempt resulting in a lot of wasted calls to the Z3-solver, we propose an invariant ranker (iRank) based on the LLM-based embedding models (i.e., `text-embedding-ada-002`, and `davinci-similarity`) and contrastive learning.  We empirically show that after generating a set of invariants with LLM, if we reranked them using $iRank$, the number of Z3 calls are significantly reduced.
>     3. We create a ranking dataset, containing the invariant synthesis problem, LLM-generate invariant candidates, all the SMT-2 (SyGus) format.
>     4. We open source our dataset, code, and trained model for future research.
> We will add a subsection at the end of the introduction to summarize these contributions.
>
> 2. **Clarification of Figure1b:** Thanks for the suggestion. Here is the figure we will add to the camera-ready version to articulate iRank’s architecture. We have uploaded the redrawn image [in this anonymous link](https://figshare.com/ndownloader/files/42163257/preview/42163257/preview.jpg).
>
>     For the embedding transformation, we use a 3-layered fully connected feedforward ANN. The hidden dimension of the embedding transformation model depends on the initial embedding model. For instance, when we use embeddings from `text-embedding-ada-002` model, the hidden dimension is $1536$. For `davinci-similarity` model, the hidden dimension is $12288$. Also note that, since we are extracting initial embeddings through OpenAI API, the only learnable parameters in $iRank$ are in the Embedding Transformation module.
> 3. **Training Details:** Apologies for the confusion. We use two different OpenAI embedding models (`text-embedding-ada-002` and `davinci-similarity`) to extract the initial embeddings for both the problem definition and the invariant candidates using the iRank architecture described above. We optimize iRank to bring the transformed embedding of the problem closer to the correct invariant and further away from the incorrect invariant from the training set. To optimize, we use the `MSELoss` where the expected similarity of the problem and correct invariant is $1$, and with incorrect invariants, it is $0$. Here are the training details.
>     ```json
>     {
>         "optimizer": "Adam",
>         "learning_rate": 5e-05,
>         "num_train_epochs": 20,
>         "weight_decay": 0.001,
>         "max_grad_norm": 1.0,
>         "lr_scheduler_type": "linear",
>         "warmup_steps": 500,
>         ...
>     }
>     ```
>     We pass the training parameters as a user input and pass these parameters as a json configuration to the training. We release the default configuration (including these values mentioned above) as part of our artifact release.  We evaluate iRank based on a $5$-fold cross validation. In each of these $5$ experiments, the set of problems used in training and in evaluation are completely separate.
>
> 4. **Complexity Analysis:**
>     1. We agree with the reviewer that the dataset is small (we have only 541 problems ). However, our results demonstrate that, even with such a small dataset, the ranker learns the semantics of the invariant synthesis problem, i.e., iRank can place the correct invariant on top of majority of the incorrect invariants. We believe with additional problems (and corresponding correct and incorrect invariants) added to the training data, performance of iRank will increase.
>     2. For training $iRank$, the maximum training epoch is set to $20$. On average, it takes iRank training $9$ epochs to reach maximum validation metric (which in our case is $V@1$). This shows, even with this small data, iRank is not learning any trivial ranking knowledge (our hypothesis is that if $iRank$ was learning trivial ranking knowledge, the maximum validation metric would’ve been reached very soon after the start of the training). We performed another experiment with TF-IDF based reranker, and the performance of such is very poor compared to iRank. For ranking `gpt-3.5-turbo` generated invariant candidates, $V@1$ for TF_IDF is $17.6$, which is higher than expected ranks from LLM ($V@1 = 8.8$). However, $V@1$ is $28.0$ and $29.2$, for $iRank-ada$, and for $iRank-davinci$, respectively. For ranking `gpt-4` generated invariant candidates, $V@1$ for TF_IDF is $32.0$, which is higher than expected ranks from LLM ($V@1 = 23.6$). However, $V@1$ is $33.6$  and $33.5$, for $iRank-ada$, and for $iRank-davinci$, respectively.
>     3. During ranking, iRank computes the dot product of the problem embedding and all the candidate invariant embeddings. If the number of candidate invariants is $O(n)$ and the embedding dimension is $d$, the complexity of reranking is $O(n*d) + \Theta(nlgn)$ (for sorting).
>
> We really hope these answers clarify any confusion. Once again, we thank you for the review and the comment. We will add these in the camera-ready version of the paper.

---

### Official Review · Reviewer_ZsSr · 2023-08-05

**Soundness:** 4

**Excitement:**

4: Strong: This paper deepens the understanding of some phenomenon or lowers the barriers to an existing research direction.

**Paper Topic And Main Contributions:**

In this submission, the authors consider the generation of loop invariants, which are essential in deductive program verification. They propose a re-ranking for LLM-generated loop variants. Although finding correct loop invariants can be done by guessing and checking, they argue that ranking the generated invariants is important to minimize the computational cost. Their approach called iRank, uses ideas from contrastive ranking: The basic idea (page 3) is to transform the problem embedding x to x′ = Ψ(x|θ), and transform invariant embedding y to y′ = Ψ(y|θ) and then maximize and minimize the similarity between x ′ and y ′ respectively. They create a training dataset and evaluate their ranking methodology, which results in a significant improvement in “V@K” (verified after trying k invariants).


**Questions For The Authors:**

- Are there straight-forward human heuristics to test your approach against?

**Reasons To Accept:**

- Important and aspiring research direction to use LLMs for verification tasks
- Their approach is a significant contribution, a perfect-size short paper
- Their dataset is interesting and can be used for interesting future research
- The experimental results are showing a significant performance compared to not using a ranking

**Reasons To Reject:**

- A more interesting baseline is missing, for example, some trivial rankings based on human-created or symbolic heuristics for ranking
- It is not immediately clear how challenging and interesting the verification problems are from reading the paper

**Reproducibility:**

4: Could mostly reproduce the results, but there may be some variation because of sample variance or minor variations in their interpretation of the protocol or method.

**Reviewer Confidence:**

4: Quite sure. I tried to check the important points carefully. It's unlikely, though conceivable, that I missed something that should affect my ratings.

---

> ### Author Rebuttal · Authors · 2023-08-29
>
> We thank the reviewer for thoughtful and valuable feedback and suggestions. Following the suggestion from the reviewer, we conducted several experiments. Here are the details.
>
> 1. **Other baselines**
>
>     - We implemented a trivial similarity-based (TF-IDF) ranker, where the invariant exhibiting higher similarity with the input problem goes to higher ranks. Since the problem definition contains the SMT formula for the `precondition` `transfer-function` and `postcondition`. This experiment tests a hypothesis _an invariant that takes most of its ingredients (i.e., SMT clauses) from the input problem has a higher chance of being successful_. We find that TF_IDF does better than LLM-ranks and even better than the raw embedding from OpenAI embedding models. Nevertheless, iRank performance is still better than all other baselines. For ranking `gpt-3.5-turbo` generated invariant candidates, $V@1$ for TF_IDF is 17.6, which is higher than expected ranks from LLM ($V@1 = 8.8$). However, $V@1$ is $28.0$ and $29.2$, for $iRank-ada$, and for $iRank-davinci$, respectively. For ranking `gpt-4` generated invariant candidates, $V@1$ for TF_IDF is $32.0$, which is higher than expected ranks from LLM ($V@1 = 23.6$). However, $V@1$ is $33.6$  and $33.5$, for $iRank-ada$, and for $iRank-davinci$, respectively.
>     - Further, we experimented on a symbolic heuristic where we retained semantically unique invariants generated by the LLMs. We evaluate whether two invariant $i_1$ and $i_2$ semantic equivalent by evaluating the following SMT formula with z3, $\forall{v_1, v_2, …, v_n}(i_1(v_1, v_2, …, v_n) \Leftrightarrow i_2(v_1, v_2, …, v_n))$. We find that GPT3.5 generated $9.29$% semantically duplicate invariants, and GPT-4 generated $4.70$% semantically duplicate invariants. We calculate the expected ranks of such a deduplicated dataset. Below is the result when ranking the invariant candidates generated by `gpt-3.5-turbo`.
>         | Experiment                   | Mean $i^+$ rank|  Median $i^+$ rank | $V@1$ |  $V@5$ |  $V@10$ |
>         |------------------------------|------------------|---------|---------|------|-------|
>         | LLM-Ranks                    | $189.78$           | $62.00$   | $5.2$     | $11.6$ | $18.4$  |
>         | Expected LLM Ranks           | $95.35$            | $31.02$   | $8.0$    | $19.2$ | $25.2$  |
>         | Expected after Deduplication | $65.24$            | $24.07$   | $8.4$     | $22.8$ | $31.2$  |
>         | iRank-ada                    | $38.78$            | $5.00$    | $28.0$    | $51.2$ | $60.8$  |
>         | iRank-davinci                | $34.48$            | $4.00$    | $29.2$    | $52.8$ | $62.8$  |
>
>         In the case of  invariant candidates generated by `gpt-4`, we find the following results,
>         | Experiment                   | Mean $i^+$ ranks | Median $i^+$ ranks | $V@1$ | $V@5$ | $V@10$ |
>         |------------------------------|------------------|--------------------|-------|-------|--------|
>         | LLM-Ranks                    | $39.20$            | $9.00$               | $17.6$  | $40.4$  | $51.6$   |
>         | Expected LLM Ranks           | $20.23$            | $4.96$               | $31.9$  | $52.1$  | $65.4$   |
>         | Expected after Deduplication | $13.99$            | $4.89$               | $31.4$  | $53.7$  | $72.8 $  |
>         | iRank-ada                    | $13.18$            | $2.00$               | $44.7$  | $74.4$  | $81.4$   |
>         | iRank-davinci                | $11.96$            | $2.00$               | $44.7$  | $71.8$  | $81.9$   |
>
>         For the `gpt-3.5-turbo` generated invariant, the mean and median expected ranks of the correct invariant are $65.24$ and $24.07$, respectively. In contrast, the expected rank in the original generation (without deduplication) were $95.35$ and $31.02$ (mean and median). For the case of `gpt-4`, the mean expected rank reduced from $20.23$ to $13.99$, and the median reduced from $4.96$ to $4.89$. So, we do see some improvements in ranks with semantic heuristics. However, such deduplication is very expensive, since we must compute $n*n$ equivalence check queries to Z3. For our experiment, it took more than $23$ hours just to deduplicate. Given the resource expense, we conjecture that such deduplication-based ranking is not feasible for real-world applications. In addition, we evaluated $iRank$ on this semantically deduplicated candidate invariant set and found that $iRank$  improves the rank of correct invariants, even when semantic duplicates are eliminated - supporting that $iRank$ is not learning any trivial semantic patterns.
>
> 2. **Complexity of the problems:** We evaluate iRank on the SyGus benchmark, which includes the loop invariant synthesis problem from several different sources, including different verification challenges. The problems include _Linear Integer Arithmetic_ ($91.68$%, $496$ out of $541$), _Array Linear Integer Arithmetic_ ($3.33$%, $18$ out of $541$) and _Non-linear Integer Arithmetic_ ($4.99$%, $27$ out of $541$) problems. Here are more detailed statistics of the problems in the dataset.
>
>     | Problem Type Statistics         |                |
>     |---------------------------------|----------------|
>     | Linear Integer Arithmetic       | $496$ ($91.68$%) |
>     | Non-Linear Intehger Arithmetic  | $29$ ($4.99$%)   |
>     | Array Linear Integer Arithmetic | $18$ ($3.33$%)   |
>
>     | Problem Semantics Statistics |                          |
>     |------------------------------|--------------------------|
>     | Number of functions          | Min = $3$, Max = $9$     |
>     | Number of variables          | Min = $2$, Max = $90$    |
>     | Variable Types               | Integer = $80.31$%       |
>     |                              | Boolean = $19.31$%       |
>     |                              | Array[Integer] = $0.36$% |
>     |                              | Array[Boolean] = $0.03$% |
>
>     | Operator Statistics (of correct invariants)        | % of all operators | Avg. # per examples |
>     |----------------------------------------------------|--------------------|--------------------------|
>     | Conjuctions                                        | $43$                 | $1.12$                     |
>     | Disjunctions                                       | $20$                 | $0.53 $                    |
>     | Negations                                          | $37$                 | $0.96$                     |
>     | Addition/Subraction                                | $43$                 | $0.5$                      |
>     | Multiplication/Division                            | $3.1$                | $0.18$                     |
>     | Logical Comparison                                 | $88.5$               | $2.25$                     |
>     | Examples with quantifiers ($\forall$ or $\exists$) | $7$                  |
>
>     | Length Statistics           | Min | Max  | Avg |
>     |-----------------------------|-----|------|-----|
>     | Problem Length (NLTK)       | $92$  | $2732$ | $740$ |
>     | Problem Length (TokToken)   | $88$  | $3146$ | $922$ |
>     | Invariant Length (NLTK)     | $18$  | $605$  | $110$ |
>     | Invariant Length (TikToken) | $15$  | $506$  | $125$ |
>
> We appreciate the valuable review and the feedback. We will incorporate our responses to the feedback to the camera-ready version of the paper.

---

### Meta-Review · Area_Chair_AyCN · 2023-09-17

**Recommendation:** 4

**Metareview:**

The authors propose a new method for synthesizing inductive loop invariants in program verification, as well as an accompanying dataset for future advancements in this direction. Reviewers agree the approach offers new speed ups and analysis will be useful to practitioners facing this problem. Listed limitations/concerns from the reviewers mainly concentrate around discussion of how this work's contributions differ/advance prior work, and qualitative analysis of iRank.

---

### Decision · Program_Chairs · 2023-10-07

**Decision:**

Accept-Findings

**Comment:**

The authors propose a new method for synthesizing inductive loop invariants in program verification, as well as an accompanying dataset for future advancements in this direction. Reviewers agree the approach offers new speed ups and analysis will be useful to practitioners facing this problem. Listed limitations/concerns from the reviewers mainly concentrate around discussion of how this work's contributions differ/advance prior work, and qualitative analysis of iRank.